# Exploring physical activity and patient perceptions in knee osteoarthritis: A mixed-methods study

Moayad Subahi⬤*, Fahda Alshaikh, Eyad Dahlawi⬤, Feras Zafar, Tamim Alsulimany, Nawaf Alnefaie, Abdulrahman Almalki

Medical Rehabilitation Sciences Department, Faculty of Applied Medical Science, Umm Al-Qura University, Mecca, Saudi Arabia

* mssubahi@uqu.edu.sa

## Abstract

Knee osteoarthritis (KOA) is a prevalent condition that reduces physical function and quality of life. Physical activity is foundational to KOA management; however, patient engagement and perceptions of treatment remain underexplored, particularly in Middle Eastern populations. This study evaluated physical activity (PA) levels among individuals with KOA and explored their perceptions, awareness, and experiences with management strategies, especially physical therapy. A sequential explanatory mixed-methods design was employed. Quantitative data were collected using the International Physical Activity Questionnaire-Short Form (IPAQ-SF) from 60 adults with physician-diagnosed KOA (mean age 55.5±6.4 years; 50% female) recruited from clinics and community programs in Saudi Arabia. Semi-structured interviews with 24 purposively selected participants further explored experiences and perceptions. Descriptive statistics summarized quantitative data, and thematic analysis guided qualitative interpretation. Ninety percent of participants recorded low PA levels (≤600 MET-min/week), with only 10% achieving moderate or high activity levels. Qualitative themes revealed multiple barriers including emotional distress, logistical difficulties, and misconceptions about KOA as well as facilitators such as family support and patient education. Integration of findings highlighted how contextual and psychosocial factors influence PA engagement. Adults with KOA in this study reported markedly low levels of PA, shaped by cultural, psychological, and environmental factors. Our findings highlight the importance of addressing these barriers through patient-centred education and improved access to physical therapy.

## Introduction

Knee osteoarthritis (KOA) is a chronic degenerative joint disease characterized by progressive loss of articular cartilage, subchondral bone changes, and functional

**Data availability statement:** The anonymized quantitative dataset is available on request from the Local Committee for Biological and Medical Ethics at Umm Al-Qura University, contact (gssr@uqu.edu.sa). The qualitative transcripts cannot be shared publicly due to ethical restrictions, but illustrative excerpts are provided in the manuscript.

**Funding:** The author(s) received no specific funding for this work.

**Competing interests:** The authors have declared that no competing interests exist.

decline [1,2]. It is a leading cause of pain and disability worldwide, with global prevalence estimates ranging from 16% in adults over 45 years to more than 22% in those aged 60 years and older [3]. In Saudi Arabia, recent community-based studies report KOA prevalence rates of approximately 30–40% among older adults, reflecting a major public health concern [4,5]. The etiology of KOA is multifactorial, involving mechanical overload, obesity, genetic predisposition, and inflammatory processes, which together contribute to joint degeneration and pain [6,7].

Contemporary management of KOA emphasizes non-pharmacological strategies—particularly education, weight management, and exercise/physical therapy—alongside pharmacological options and, when indicated, surgery, in line with recent international guidelines (e.g., NICE, EULAR, OARSI). Choice of treatment should reflect disease severity, comorbidities, and patient preferences, and prioritize sustained physical activity and self-management support [8–10].

Physical therapy is essential in managing KOA, improving joint mobility, strengthening the muscles around the knee, and educating patients on self-management techniques [11]. Several studies have demonstrated that physical therapy can significantly reduce pain and enhance joint function in patients with KOA [12,13]. Modalities such as therapeutic exercises, manual therapy, heat therapy, electrical stimulation, and assistive devices are key components of effective physical therapy interventions [14,15]. Strengthening exercises, in particular, are essential in these programs, targeting muscles around the knee joint to promote stability, range of motion, and cardiovascular fitness [16].

Physical activity (PA) is also integral to KOA management, reducing pain, improving joint function, and enhancing quality of life [17,18]. Despite its importance, many individuals with KOA remain insufficiently active due to barriers such as pain, fear of exacerbating symptoms, and a lack of motivation or accessible exercise facilities [19,20]. Studies emphasize the need for tailored interventions to address these challenges, including educational programs, guided exercise plans, and support from healthcare providers [21,22]. By overcoming these barriers, patients can engage in regular PA, which is crucial for slowing disease progression and improving long-term outcomes [23].

In Saudi Arabia, lower public awareness of KOA including limited understanding of mechanisms, risk factors, and evidence-based care has been repeatedly documented. A study found that only 37% correctly identified core KOA mechanisms, with lower educational attainment associated with poorer knowledge [24]. Similar patterns were observed in another study, where most respondents demonstrated poor or fair awareness [25]. These findings indicate a graded education–awareness relationship and highlight the need for targeted, culturally sensitive education campaigns (e.g., clinics, community programs, and vetted digital platforms) to counter misconceptions and support earlier, active management.

Addressing psychosocial burden is critical: people with KOA frequently report anxiety, depressive symptoms, fear of movement, and reduced self-efficacy, all of which are associated with lower activity and worse function [26–29]. Education, family support, and therapeutic alliance can mitigate these factors and improve

engagement with activity-based care [30,31]. Integrating psychosocial support within multidisciplinary management may therefore facilitate adherence to recommended exercise and daily physical activity [32]. Healthcare providers may also face challenges managing KOA due to patients' lack of awareness and misconceptions about the disease. Studies maintained that uncertainty among clinicians results from inconsistent treatment strategies and patient distrust [33–35]. This highlights the importance of providing comprehensive education to patients throughout the conservative treatment process. Additionally, fostering trust and clear communication between patients and providers is essential for improving adherence to therapy and achieving better outcomes [36]. The literature reveals significant gaps in awareness and understanding of KOA, particularly in Saudi Arabia, which hinder effective management and highlight the need for targeted educational efforts. These efforts should aim to empower patients with knowledge, address physical and psychological barriers, and promote evidence-based practices to mitigate the burden of KOA while paving the way for improved multidisciplinary care.

Despite guideline consensus on the centrality of physical activity and physical therapy, context-specific evidence from Middle Eastern settings remains limited, particularly regarding how cultural, social, and psychological factors shape engagement with activity-based care. This study therefore employed a sequential explanatory mixed-methods design to (i) quantify PA levels among adults with KOA and (ii) explore their perceptions, awareness, and experiences of KOA physical therapy management in Saudi Arabia.

## Materials and methods

### Study design and setting

This study employed a sequential explanatory mixed-methods design, consisting of two consecutive phases: (i) a quantitative survey of PA levels using the International Physical Activity Questionnaire–Short Form (IPAQ-SF), followed by (ii) a qualitative phase using semi-structured interviews to explore patient perceptions and experiences. The design was chosen to enable statistical description of PA levels and to provide deeper explanatory insights into the quantitative findings through thematic analysis of interview data. The study was conducted between 25/Oct/2023 to 25/Feb/2024 in Saudi Arabia.

### Research paradigm and reflexivity

The study was underpinned by a pragmatic paradigm, which guided the use of both quantitative and qualitative methods to best address the research objectives. Pragmatism emphasizes the research question as central and allows for methodological pluralism, enabling integration of numerical data with patient narratives to generate contextually relevant insights [37]. Reflexivity was maintained throughout, with the research team consisting of physiotherapists and academic researchers trained in musculoskeletal rehabilitation [38]. To minimize bias, interviewers acknowledged their clinical background, used open-ended questions, and engaged in peer debriefing to reflect on how their perspectives might influence data interpretation. Reporting adhered to COREQ recommendations [39].

### Participants

**Quantitative phase.** A total of 60 adults with KOA were recruited through purposive and convenience sampling from physiotherapy clinics and community programs. Inclusion criteria included adults aged 45 years or older with a physician-confirmed diagnosis of KOA, as per radiographic or clinical evaluation, who could provide informed consent and participate in the assessments. Participants were required to have been under OA management for at least six months to ensure sufficient experience with treatment modalities.

**Sample size.** We did not perform an a-priori power calculation because the quantitative phase was exploratory and designed to provide descriptive estimates to inform the qualitative phase. A target of at least 60 participants was set,

which provides acceptable precision for proportions and continuous summaries in clinic-based KOA samples. This sample size was also sufficient to support exploratory subgroup contrasts (e.g., by sex) and to guide purposive selection for the qualitative interviews.

**Qualitative phase.** From the survey participants, 24 were purposively selected from the quantitative cohort using maximum variation sampling to ensure diversity in age, gender, and PA levels. Semi-structured interviews were conducted until data saturation was reached, defined as the point when no new themes emerged across three consecutive interviews. No prior relationships existed between the researcher and participants.

Exclusion criteria for both components included individuals with cognitive impairments, coexisting neurological disorders, or other severe comorbidities that could interfere with participation. Participants who agreed to participate received the participant information sheet and consent form, completed before the data collection began. Descriptive data of the 24 qualitative participants, including age, gender, BMI category, and number of physiotherapy sessions, were summarized to provide contextual understanding. Each participant was assigned a code (e.g., P01, P02) used when quoting in the results to maintain anonymity while ensuring traceability of responses.

## Quantitative data collection

Physical activity was measured using the IPAQ-SF [40], a validated tool for estimating weekly activity in METS-min/week [41,42]. The IPAQ-SF records frequency and duration of walking, moderate activity, vigorous activity, and sitting over the previous 7 days. Responses were converted into metabolic equivalent minutes per week (MET-min/week) using the IPAQ scoring protocol. Participants were categorized as having low (≤600 METS-min/week), moderate (600–3000 METS-min/week), or high (>3000 METS-min/week) PA levels according to the international cut-offs and threshold that has been reported in regional studies [43–47]. Data were collected before face-to-face qualitative interviews or via structured phone interviews.

## Qualitative data collection

Semi-structured interviews were conducted with 24 participants in Arabic to explore their experiences, perceptions, and barriers to OA management. An interview guide with open-ended questions was used, developed by the research team based on literature and clinical experience [25,47,48]. Interviews were conducted in Arabic by trained physiotherapists with qualitative research experience. Interviews were conducted in person or by phone, depending on participant preference, and lasted between 30–60 minutes. All interviews were audio-recorded with consent, transcribed verbatim, and translated into English by a bilingual researcher, and a second bilingual team member checked translations for accuracy and anonymized before analysis.

To minimize bias, interviewers reflected on their clinical background and relationship to participants prior to and during interviews. Reflexivity and transparency were maintained in line with the COREQ checklist. Interviews continued until thematic saturation was reached, defined as no new themes emerging across three consecutive interviews [49].

## Data analysis

Quantitative data were analyzed in SPSS version 20 (IBM Corp., Armonk, NY, USA). Descriptive statistics (means and standard deviations for normally distributed variables; medians and interquartile ranges for skewed variables) were used to summarize participant demographics and PA levels. Categorical variables were summarized as frequencies and percentages. 95% confidence intervals (CIs) were calculated for key estimates. In addition, exploratory comparisons were performed to examine potential associations between PA levels and participant characteristics. Mann–Whitney U tests were used for sex-based differences, $\chi^2$ tests for categorical variables (e.g., PA category by sex or education), and Spearman's rank correlation for continuous relationships (e.g., PA levels with BMI or age). Given the study's exploratory design, emphasis was placed on effect sizes and CIs, and p-values were interpreted cautiously.

The qualitative data were analyzed using thematic analysis following Braun and Clarke's six-step approach [50]. Transcripts were imported into NVivo 12 (QSR International, Melbourne, Australia) for management and coding. An inductive coding strategy was applied: initial open codes were generated from participants' narratives, codes were grouped into broader categories, and final themes were developed through iterative comparison. Two members of the research team independently coded a subset of transcripts, and discrepancies were resolved through discussion to enhance reliability.

Credibility was supported through peer debriefing, triangulation with quantitative findings, and reflexive memo writing throughout analysis. The research team, consisting of physiotherapists and academic researchers, reflected on their professional background and potential influence on data interpretation. Transcripts were read and re-read for familiarization, and initial codes were generated inductively, focusing on semantic and latent content. These codes were then collated into themes, which were reviewed, refined, and discussed among the research team until consensus was reached.

Themes emerged inductively from the data, reflecting participants' lived experiences. Each theme was illustrated with quotes linked to individual participants using pseudonyms (e.g., P01, female, age 55). Reflexivity was maintained throughout the process, with researchers documenting assumptions and reflecting on their influence on data interpretation [51]. As a team of rehabilitation scientists and physiotherapists based in Saudi Arabia, we acknowledged our expertise in physical therapy and cultural familiarity (e.g., prayer practices), and actively sought diverse perspectives to mitigate potential bias and ensured that thematic analysis remained grounded in participants' own words rather than preconceived assumptions. Interviews continued until thematic saturation was achieved.

The trustworthiness of the qualitative findings was ensured through targeted strategies [52]. Credibility was enhanced by triangulating qualitative themes with quantitative physical activity data from the IPAQ-sf, validating participants' experiences. Dependability was achieved by documenting Braun and Clarke's six-step thematic analysis process, providing a clear audit trail. Confirmability was supported by having two researchers independently code transcripts and resolve discrepancies, minimizing bias. Transferability was addressed through detailed descriptions of the 24 diverse participants and the Saudi Arabian context, enabling applicability assessments. These steps ensured the findings' rigour and authenticity. Reporting followed the COREQ checklist to ensure transparency and comprehensiveness [39].

### Ethical considerations

Ethical approval for the study was obtained from the Local Committee for Biological and Medical Ethics at Umm Al-Qura University (NJDD011023). All participants provided written informed consent before participating. Participants who could not attend face-to-face interviews, a verbal consent was recorded during the call. Confidentiality and anonymity were ensured through secure data handling and coding systems. Participants could withdraw at any time without impacting their medical care or legal rights, with their data promptly destroyed upon withdrawal.

## Results

### Quantitative results

**Descriptive statistics.** The study aimed to evaluate physical activity levels among individuals with KOA and explore their perceptions, awareness, and experiences with treatment options, particularly physical therapy. Sixty participants were included (50% female; mean age 55.5±6.4 years, range 45–65). The mean BMI was 27.8±5.8 kg/m2 (range 19.5–47.3). Moreover, participants attended an average of 10.8 physiotherapy sessions (SD=5.7), ranging from 3 to 20 sessions. Participants demographics summarised in Table 1.

**Physical activity level.** Physical activity levels using the standard IPAQ-SF thresholds shows that the majority of participants were classified as Low PA, with smaller proportions achieving moderate or high activity. Exact counts and percentages are presented in Table 2. Moreover, comparisons of PA levels by sex, BMI, and education level showed no statistically significant differences. The distribution of PA categories did not differ between men and women ($\chi^2$=3.30, p=0.19), and Spearman's correlation between BMI and MET-min/week was not significant ($\rho$=–0.16, p=0.23). Education

**Table 1. Participant characteristics (n = 60).**

| Variable | Mean ± SD/ n (%) |
|---|---|
| Age (years) | 55.5 ± 6.4 (range: 45–65) |
| Gender | Male: 30 (50.0%) |
| | Female: 30 (50.0%) |
| BMI (kg/m²) | 27.8 ± 5.8 |
| | Normal: 21 (35.0%) |
| | Overweight: 22 (36.6%) |
| | Obese: 17 (28.3%) |
| Education | Bachelor's degree: 17 (45.0%) |
| | Diploma: 12 (20.0%) |
| | Master's degree: 10 (16.7%) |
| | High school: 8 (13.3%) |
| | PhD: 3 (5.0%) |

Values are presented as mean ± SD for continuous variables and n (%) for categorical variables.

**Table 2. Physical activity level results.**

| Physical activity level | n (%) |
|---|---|
| Low (≤600 METS-min/week) | 54 (90%) |
| Moderate (600–3000 METS-min/week) | 4 (6.6%) |
| High (>3000 METS-min/week) | 2 (3.3%) |

level was also not associated with PA category ($\chi^2 = 5.34$, $p = 0.72$). These analyses are considered exploratory and should be interpreted with caution given the sample size.

## Qualitative results

The qualitative study resulted in six themes developed through the thematic analysis: The beginning of the patient's journey, Treatment, Providing care for individuals with KOA, The patient's therapeutic culture, Patient environment, Challenges and Barriers. **Table 3**. presents the themes, sub-categories, and illustrative quotes from patient interviews. The qualitative component highlighted the underlying reasons for these low activity levels, revealing multifaceted barriers. Emotional distress, often associated with chronic pain and reduced mobility, was frequently cited as a deterrent to maintaining regular activity. Logistical challenges, such as difficulty accessing physical therapy services and a lack of transportation, further compounded these barriers. Participants also expressed limited awareness about the benefits of exercise and common misconceptions about KOA management, leading to a reliance on passive treatments and avoidance of physical activity.

Despite these barriers, the qualitative analysis also uncovered facilitators that encouraged engagement in PA. Support from family members and healthcare providers played a significant role in motivating participants to adhere to treatment plans and exercise routines. Additionally, participants who received tailored education about KOA and the role of exercise reported greater confidence in their ability to manage symptoms through activity. Those with access to consistent PT services noted significant improvements in pain, mobility, and overall quality of life, emphasizing the importance of structured, guided interventions.

Integrating the quantitative and qualitative findings, this study highlights a critical need to address the barriers to physical activity in KOA populations. While quantitative data highlight the severity of inactivity, qualitative insights provide

**Table 3. Themes, sub-categories, and illustrative quotes from patient interviews.**

| Theme | Category | Illustrative Quote |
|---|---|---|
| Theme 1: The Beginning of the Patient's Journey | *The Onset of the Disease* | "I could not bend my knee, especially during prayer; I could not sit properly in prayer." (P02, Female, 55) |
| | *Decision-Making* | "Honestly, I prefer physical therapy over surgery; surgery should be the last option." (P07, Male, 59) |
| Theme 2: Treatment | *Patient's Preference for Therapeutic Solutions* | "I prefer to do physical therapy even if it takes longer." (P05, Female, 58) |
| | *Interest in Returning to Life* | "Thank God, I continue to take one ibuprofen pill a day and one vitamin B12 pill." (P11, Male, 53) |
| | *Exercise* | "My experience with the exercises from the physiotherapist has been very satisfying. The pain has decreased a lot." (P14, Female, 60) |
| Theme 3: Providing Care for Individuals with KOA | *Physical Therapy: Perceptions and Reality* | "The doctor said the surgery was successful, but 70% of the recovery depends on physical therapy." (P01, Male, 61) |
| | *Access and Quality of Care* | "The doctor would call me online and conduct a virtual physical therapy session." (P12, Female, 57) |
| | *From Goals to Gains: Patient's Progress and Outcomes* | "After physical therapy, I felt relieved, and I was able to walk again." (P08, Male, 62) |
| Theme 4: The Patient's Therapeutic Culture | *Curiosity and Awareness: Understanding KOA* | "Is this surgery beneficial or not? Have many people been able to walk perfectly?" (P17, Female, 54) |
| | *Options and Experiences: Perspectives on Care* | "They just give brief instructions and don't follow up on the exercises." (P20, Male, 56) |
| Theme 5: Patient Environment | *Community Support: A Pillar of Healing* | "My friends, my husband, and my children help me." (P03, Female, 59) |
| | *Myths and Misconceptions: Navigating Patient Advice* | "How does physical therapy work for KOA? Can the cracking sound go away?" (P16, Male, 52) |
| Theme 6: Challenges and Barriers | *Other Diseases and Injuries* | "I had an accident 25 years ago that caused complications in my knee." (P04, Male, 63) |
| | *Obstacles and Emotions* | "Living with KOA significantly affected my mental and emotional health." (P09, Female, 55) |

a deeper understanding of patients' emotional, logistical, and educational challenges. Tailored interventions prioritizing patient education, emotional support, and accessible physical therapy services may improve PA levels and overall disease management.

**Theme 1: The beginning of the patient's journey.** The onset of KOA marked a significant disruption in participants' lives, as chronic knee pain began to impede daily activities imbued with personal and cultural significance. One participant described difficulty performing daily religious practices: *"I could not bend my knee during prayer"* (P02, Female, 55), highlighting how the condition challenged their ability to engage in religious practices central to life in Saudi Arabia. This physical limitation often triggered a cascade of emotional distress and reduced quality of life, prompting participants to seek medical advice. Consultations with orthopaedic specialists confirmed the diagnosis and introduced treatment options, with physical therapy frequently recommended as a first step. When choosing a path forward, participants leaned toward conservative approaches over surgery. As one explained, "Honestly, I personally prefer physical therapy over surgery. I'd rather avoid surgery," reflecting a common fear of invasive procedures and a hope that physical therapy could restore function less disruptively. These early experiences reveal how physical symptoms, cultural priorities, and treatment preferences shaped participants' initial steps in managing KOA, setting the stage for their engagement or lack of physical activity.

**Theme 2: Treatment.** Participants' approaches to treatment were shaped by a complex mix of preferences, practical considerations, and aspirations to reclaim their pre-OA lives. Many expressed a strong inclination toward physical therapy over surgical interventions, driven by both psychological comfort and perceived effectiveness. Many participants

expressed a strong preference for conservative treatment: *"I prefer physical therapy over surgery"* (P07, Male, 59). This highlights a deliberate choice to prioritize less invasive methods despite the time commitment, reflecting a broader reluctance to embrace surgery unless necessary. For those who adhered to treatment plans often combining medication and physical therapy the rewards were evident. Another participant noted, "Thank God, I continue to take one ibuprofen pill a day and one vitamin B12 pill," suggesting that a manageable regimen supported their persistence.

Exercise, a cornerstone of physical therapy, elicited varied responses. For some, it was transformative: "My experience with the exercises given to me by the physiotherapist at the hospital has been very satisfying. The pain has significantly decreased and improved a lot." This positive outcome highlights how tailored therapeutic routines can bridge the gap between sedentary behaviour (noted in 90% of participants with ≤600METS-min/week) and improved function. However, logistical barriers tempered the enthusiasm of others, who found attending sessions inconvenient and yearned for at-home alternatives. This tension between the benefits of exercise and the challenges of accessing it points to a critical need for flexible, patient-centred solutions to enhance adherence and activity levels.

**Theme 3: Providing care for individuals with KOA.** Participants' experiences with care delivery, particularly physical therapy, revealed a spectrum of perceptions that evolved with exposure to treatment. Initially, some believed in disbelief or confusion about its efficacy, but tangible improvements often shifted their views. One participant reflected, *"The doctor said the surgery was successful, but most recovery depends on physical therapy"* (P01, Male, 61). This insight, gained post-surgery, highlights the pivotal role of physical therapy in achieving functional outcomes, reinforcing its value beyond initial expectations. Access to quality care further shaped experiences. Innovative solutions like virtual sessions were a boon for some: "The doctor would call me online and conduct a virtual physical therapy session", offering flexibility amid logistical challenges like transportation or long wait times. However, inconsistent communication and appointment delays frustrated others, highlighting disparities in care delivery.

When physical therapy aligned with participants' goals, the gains were palpable. "After undergoing physical therapy, I felt relieved, and I was able to walk on my leg again," one participant shared, illustrating how consistent engagement could restore mobility and alleviate pain. However, missed sessions or neglected home exercises disrupted progress, emphasizing that the benefits of care hinge on sustained participation. These findings suggest that while physical therapy holds promise for improving low activity levels, its success depends on overcoming barriers to access and fostering patient-provider collaboration.

**Theme 4: The patient's therapeutic culture.** Participants' therapeutic culture of knowledge, curiosity, and attitudes toward KOA management revealed gaps and glimmers of awareness that influenced their treatment journeys. Some participants expressed uncertainty about treatment effectiveness, asking, *"Is this surgery beneficial?",* this curiosity reflects an active desire to understand their condition, yet it often coexisted with limited baseline knowledge about KOA and its management. Physical therapy, however, emerged as a point of recognition for some, who valued its role in strengthening muscles and stabilizing joints, even if their understanding was incomplete.

Experiences with healthcare systems further shaped perspectives. Dissatisfaction surfaced when care felt cursory: "They just give brief instructions and don't even follow up on some of the exercises," one participant lamented, pointing to a perceived lack of personalization in public settings. In contrast, those who received detailed explanations or accessed private care expressed greater satisfaction, suggesting that quality of interaction influences trust and engagement. Despite these challenges, hope persisted: "I believe consistent physical therapy can improve my quality of life," another noted, indicating that education and positive care experiences could empower patients to embrace evidence-based practices over time. This theme highlights a critical need for targeted education to bridge knowledge gaps and align therapeutic culture with effective OA management.

**Theme 5: Patient environment.** The patient's environment of family, friends, and community served as both a pillar of support and a source of complexity in their recovery journey. Emotional and practical encouragement from loved ones was invaluable, family support was frequently highlighted: *"My husband and children help me"* (P03, Female, 59), emphasizing

how social networks mitigated isolation and supported adherence to treatment plans. Community interactions, however, introduced a mix of helpful and misguided advice. Traditional remedies, passed down through generations, held appeal for some, yet often lacked evidence. Another participant questioned prevailing myths: "How does physical therapy work, especially for KOA? And how can cracking sound go away?" This reflects a broader struggle to navigate well-intentioned but unsupported suggestions, such as dietary fixes or unproven treatments, which sometimes affect evidence-based options.

These environmental dynamics illuminate why many participants remained sedentary. Community misconceptions reinforced avoidance of activity, while supportive networks offered a counterbalance that encouraged persistence. Addressing this duality requires education that extends beyond the individual to their social circles, ensuring that support aligns with clinical recommendations and fosters greater physical activity engagement.

**Theme 6: Challenges and barriers.** Living with KOA presented multifaceted challenges that compounded physical limitations with emotional and logistical hurdles. Coexisting conditions amplified difficulties: "I had an accident 25 years ago that caused complications in my knee," one participant explained, illustrating how past injuries or comorbidities complicated recovery and deterred activity. External disruptions, like the COVID-19 pandemic, further derailed care, forcing delays or abandonment of therapy sessions and exacerbating sedentary tendencies (reflected in the 90% low activity rate).

The emotional toll was profound, as one participant stated: *"Living with KOA affected my mental and emotional health"* (P09, Female, 55). Another participant confessed, capturing the frustration, anxiety, and isolation that stemmed from restricted mobility. This psychological burden often reinforced a cycle of inactivity, as pain and fear of worsening symptoms deterred exercise. These barriers may highlight the need for holistic interventions integrating emotional support, tailored education, and accessible care to break the sedentary pattern and improve outcomes. Participants' narratives reveal that overcoming these challenges is not just a physical attempt but a profoundly personal one, requiring a biopsychosocial approach to OA management.

## Integration of quantitative and qualitative results

The qualitative themes provided explanatory depth to the quantitative finding that 90% of participants reported low PA (<600 MET-min/week). The absence of significant associations between PA levels and sex, BMI, or education in exploratory analyses suggests that demographic factors alone do not explain the sedentary patterns observed. Instead, participants' narratives highlighted how psychosocial distress and logistical barriers (Theme 6) discouraged engagement, while misconceptions and limited awareness (Theme 4) reinforced inactivity. At the same time, supportive family environments (Theme 5) and positive experiences with physical therapy (Themes 2 and 3) facilitated motivation and adherence for a small subgroup who achieved moderate-to-high PA levels. This convergence demonstrates how personal, social, and cultural influences rather than demographic characteristics primarily shaped PA behaviour. Such integration highlights the value of a sequential explanatory mixed-methods design, where qualitative insights contextualize and expand quantitative results [53]. Collectively, the findings emphasize the importance of culturally sensitive, patient-centered strategies to address barriers and support sustained PA in individuals with KOA.

## Discussion

This mixed-methods study explored PA levels and patient experiences in KOA management in Saudi Arabia. The findings emphasize the substantial gap in adherence to PA guidelines, with 90% of participants classified as having low PA levels. This highlights a critical area for intervention in KOA care. The barriers identified in the qualitative analysis align with existing literature, emphasizing the role of emotional distress, logistical challenges, and limited awareness in hindering physical activity [54,55]. Participants frequently reported that chronic pain, reduced mobility, and misconceptions about KOA management contributed to their sedentary behaviour. These findings are consistent with global evidence highlighting fear

of pain, low confidence, and limited support as key barriers to PA in KOA [20,56]. However, context-specific themes such as prayer and reliance on family advice illustrate the cultural uniqueness of our cohort. International guidelines emphasize structured exercise, weight management, and education as first-line strategies [2,9,57], but implementation in Saudi Arabia must account for these cultural factors to improve adherence. In addition, the findings reinforce the importance of addressing psychological and educational aspects alongside physical symptoms to improve patient outcomes.

Although equal numbers of male and female were included, no significant gender-based differences in PA were observed. Studies suggested that women reported greater logistical barriers (e.g., access to facilities), while men more often emphasized fear of worsening symptoms [58,59]. Similarly, BMI and education were not significantly associated with PA, suggesting that psychosocial and cultural influences outweighed demographic factors in shaping PA behaviours.

Notably, the study highlights facilitators that can enhance PA engagement. Support from family members and healthcare providers emerged as a significant motivator, suggesting that a collaborative approach involving the patient's social network could yield positive results. Tailored education about the benefits of PA and accessible physical therapy services were also identified as key enablers of that may improve PA levels. These findings align with the biopsychosocial model of care, which advocates for a holistic approach to managing chronic conditions like KOA.

The integration of quantitative and qualitative findings provides robust evidence for the development of targeted interventions. For instance, implementing community-based education programs to address misconceptions and increase awareness about KOA management could empower patients to take an active role in their care. Similarly, expanding access to physical therapy services, particularly in underserved areas, could help overcome logistical barriers and improve adherence to exercise regimens.

Future research should focus on evaluating the effectiveness of tailored, multifaceted interventions that address the unique barriers and facilitators identified in this study. Longitudinal studies examining the impact of such interventions on physical activity levels, pain management, and quality of life in KOA patients would further strengthen the evidence base.

### Limitations

This study has several limitations that should be acknowledged. First, the sample size for the quantitative and qualitative components was relatively small, which may limit the generalizability of the findings. The study was conducted in a single geographic region, and cultural factors specific to Saudi Arabia may influence the experiences and perceptions of participants. Second, the study's cross-sectional design does not allow for causal inferences regarding the relationship between identified barriers and PA levels. Longitudinal studies are needed to explore these dynamics over time. The quantitative phase did not use an a-priori power calculation and was estimation-focused; accordingly statistical tests are interpreted as exploratory. Nevertheless, the achieved sample provided acceptable precision for the study's descriptive aims and informed the qualitative sampling and interpretation. The reliance on self-reported measures for PA levels, such as the IPAQ-sf, may introduce recall bias, potentially affecting the accuracy of the results. While our findings provide insight into Saudi adults with KOA, they may not be directly transferable to other populations without cultural adaptation.

### Conclusion

Integrating PA-level assessment with qualitative insights offers a robust understanding of the challenges and facilitators in KOA management. Addressing barriers through tailored interventions that include education, emotional support, and accessible physical therapy services may support the adherence to PA and overall quality of life in KOA patients. This study highlights the need for patient-centred approaches that consider the biopsychosocial dimensions of KOA care.

### Author contributions

**Conceptualization:** Moayad Subahi, Nawaf Alnefaie.

**Formal analysis:** Moayad Subahi, Fahda Alshaikh, Eyad Dahlawi, Feras Zafar, Tamim Alsulimany, Abdulrahman Almalki.

**Investigation:** Moayad Subahi, Eyad Dahlawi, Feras Zafar, Tamim Alsulimany, Nawaf Alnefaie, Abdulrahman Almalki.

**Methodology:** Moayad Subahi, Fahda Alshaikh, Tamim Alsulimany, Abdulrahman Almalki.

**Project administration:** Moayad Subahi.

**Supervision:** Moayad Subahi.

**Writing – original draft:** Moayad Subahi, Fahda Alshaikh, Eyad Dahlawi, Feras Zafar, Tamim Alsulimany, Nawaf Alnefaie, Abdulrahman Almalki.

**Writing – review & editing:** Moayad Subahi, Fahda Alshaikh, Nawaf Alnefaie.

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
