## [Decision Letter · Decision Letter 0]

20 Aug 2025

Dear Dr. Subahi,

Thank you for submitting your manuscript to PLOS ONE. After careful consideration, we feel that it has merit but does not fully meet PLOS ONE’s publication criteria as it currently stands. Therefore, we invite you to submit a revised version of the manuscript that addresses the points raised during the review process.

We look forward to receiving your revised manuscript.

Kind regards,

Domiziano Tarantino, MD

Academic Editor

PLOS ONE

2. In the online submission form you indicate that your data is not available for proprietary reasons and have provided a contact point for accessing this data. Please note that your current contact point is a co-author on this manuscript. According to our Data Policy, the contact point must not be an author on the manuscript and must be an institutional contact, ideally not an individual. Please revise your data statement to a non-author institutional point of contact, such as a data access or ethics committee, and send this to us via return email. Please also include contact information for the third party organization, and please include the full citation of where the data can be found.

Reviewers' comments:

Reviewer's Responses to Questions

**Comments to the Author**

1. Is the manuscript technically sound, and do the data support the conclusions?

Reviewer #1: Yes

Reviewer #2: Partly

Reviewer #3: Partly

Reviewer #4: Partly

Reviewer #5: Partly

Reviewer #6: Partly

Reviewer #7: Yes

2. Has the statistical analysis been performed appropriately and rigorously?

Reviewer #1: No

Reviewer #2: Yes

Reviewer #3: No

Reviewer #4: No

Reviewer #5: Yes

Reviewer #6: I Don't Know

Reviewer #7: Yes

3. Have the authors made all data underlying the findings in their manuscript fully available?

Reviewer #1: No

Reviewer #2: Yes

Reviewer #3: Yes

Reviewer #4: Yes

Reviewer #5: No

Reviewer #6: Yes

Reviewer #7: Yes

4. Is the manuscript presented in an intelligible fashion and written in standard English?

Reviewer #1: Yes

Reviewer #2: No

Reviewer #3: Yes

Reviewer #4: Yes

Reviewer #5: Yes

Reviewer #6: Yes

Reviewer #7: Yes

Reviewer #1: Major concern

• Lack of Theoretical Framework, Research Paradigm, and Researcher Positionality

The qualitative component does not clearly state the underlying theoretical framework or research paradigm guiding the analysis (e.g., constructivist, interpretivist). This omission limits understanding of the epistemological stance taken in the study. Additionally, while the authors briefly mention that reflexivity was maintained, they do not provide a clear account of researcher positionality—such as their professional backgrounds, potential biases, or their influence on data collection and interpretation. These elements are essential for ensuring transparency, credibility, and rigor in qualitative research and should be explicitly reported, ideally with reference to qualitative reporting standards (e.g., COREQ).

• Sample Size Justification and Generalizability

The manuscript does not provide a justification for the sample sizes used in either the quantitative (n = 60) or qualitative (n = 24) components. No power analysis or reference to prior research is provided to support the adequacy of the quantitative sample.

• Integration of Findings

The mixed-methods integration could be strengthened by more explicitly linking quantitative and qualitative results. For instance, connecting illustrative quotes to participants’ physical activity classifications would improve the depth of interpretation and the alignment between datasets.

• Limited Analytical Depth in Quantitative Analysis

The quantitative component is restricted to descriptive statistics. Incorporating inferential analyses (e.g., exploring relationships between physical activity levels and demographic or treatment variables) would enhance the rigor and clinical relevance of the findings.

Recommendation: Major Revision

The study addresses a relevant topic, but key methodological issues—particularly in qualitative framework with paradigm used, sample justification, and data integration—need to be addressed before it is suitable for publication.

Reviewer #2: inconsistent capitalization in themes

misses recent global guidelines on OA management (e.g., NICE, EULAR 2023 updates)

There is no justification for the sample size—especially the choice of 60 for IPAQ, which could affect statistical validity

Please try clearer alignment between each qualitative theme and corresponding quantitative data

Reviewer #3: What was the reason for conducting your mixed methods study?

Explain the order of quantitative and qualitative stages of the study.

Did you take any specific action or was it just a descriptive study?

Was the quantitative instrument validated and reliable?

Was sample size determined for the quantitative part of the study?

Do not repeat the data in the table in the text.

A table of participant characteristics should be provided.

Please indicate which participant the quotations were from.

The discussion needs serious reinforcement. Compare the study results with similar studies.

Reviewer #4: Dear authors,

Your article finds the scope and aim of Plos One, offering a concise and comprehensive overview of knee OA.

I appreciate your work and would like to recommend improving the following aspect that could be refined in your manuscript PONE-D-25-20426.

Abstract

a. Please insert demographic information – these are critical for understanding the study context and allow generalizability (rows 20, 21).

b. Also, the setting and recruitment strategy are not mentioned in the abstract.

Introduction

a) I suggest including in this section prevalence data and etiology.

b) Since the topic focuses either on emotional distress, I recommend inserting in this section how KOA affects the subject and which areas are influenced by this pathology (eg, anxiety, fear, lower confidence, etc).

c) It is a poor understanding of the link that was used between education level and awareness of OA- row 62-57. I suggest reformulation, because the current phrasing doesn’t draw a direct connection that reflects “these studies highlight the need for targeted education campaigns, public lectures, and other methods to enhance community knowledge about knee OA”- row 68-69.

d) I highly recommend you also study this recent research: https://doi.org/10.3390/life14111379, https://doi.org/10.3390/life15050721, https://doi.org/10.3390/app14072886

,Materials and Methods

The study design and setting are well-structured.

Please take into consideration the recommendation about the Participants section:

a. How was the selection method for the 60 subjects?- row 108. Please mentioned.

b. I strongly suggest reorganizing this section to improve flow and clarity. For example, row 111 constitutes an additional inclusion criterion: a subject with OA who had at least 6 months of therapy. Immediately afterward, the text shifts to information about the qualitative component, which is interrupted by details about the exclusion criteria. Subsequently, information about the qualitative participants is presented again.

Please take into consideration the recommendation about the Quantitative Data Collection:

a. Since the abbreviation is IPAQ-sf, please add the full name of the tool “-Short Form”- row 129

b. Please describe the unit and formula of METS- min/ week- row 130.

c. I suggest inserting “PA level” 131-132, close to “as having low”- row 130, to ensure accuracy of information.

d. Are there some gaps in the intervals of interpretation?

≤438 METS min/week= low PA;

1100–3500 METS-min/week = moderate PA;

>4500 METS-min/week= high PA.

What happens when the patient registers values that are between these ranges? E.g., 439 and 1099 or 3501 and 4500?

Please take into consideration the recommendation about the Qualitative Data Collection:

a. Please avoid the repetition- row 137-138 and 142-142.

b. Please mention who did the translation from Arabic to English.

Additionally, please specify who conducted the interview and how the bias was mitigated.

c. The ideal for this type of study design was to use a standard questionnaire. For instance, I strongly recommend adding sourcing that supports the claim that the interview guide was based on literature, as noted in row 138.

Please take into consideration the recommendation about the Data Analysis:

a. Please add details about descriptive statistics to inform readers better (eg, means, medians, SD).

b. Please mention the version of NVivo software.

Please consider adding sources that support the trustworthiness, rows 167-177. To support this, I suggest adding references that demonstrate the reliability and validity of the COREQ and IPAQ-SF. Additionally, please provide clarity on this paragraph by adding information that addresses the following question: a) How were the qualitative themes triangulated with quantitative data? Was it through comparison, integration, or cross-validation?

Result- quantitative results

a. Suggest adding min- max – row 185;

b. Before categorizing the type of obesity (row 196), I strongly suggest inserting the BMI values in the materials and methods.

c. Regarding the Physical activity level- please be more specific with the number of participants who demonstrated moderate or high PA level- rows 209-210.

Result- qualitative results

a. Please add references to support the affirmation inserted in rows 222-227.

b. Please ensure that results support the statement inserted in rows 232- 238. How was the correlation made?

For example, patients who receive tailored education about OA reported confidence in their ability to manage symptoms, which results support this statement. And so on.

Integration of quantitative and qualitative results

A strong mixed-methods article must include statistical analysis to support and validate the quantitative part. Statistical results do not support the correlations.

1. Ahmed, Amina & Pereira, Lucas & Jane, Kimberly. (2024). Mixed Methods Research: Combining both qualitative and quantitative approaches.

Also, since it is mentioned that it was used the Triangulation Design- according to “This design is used when a researcher wants to compare and contrast quantitative statistical results with qualitative findings directly or to validate or expand quantitative results with qualitative data”.

2. John W. Creswell, Vicki L. Plano Clark. Designing and Conducting Mixed Methods Research. Chapter 4.

Discution

The Discussion section lacks references.

General recommendations:

1. Replacing OA, which is the general term for osteoarthritis, with KOA.

2. Improving the Introduction, Method and Materials, Discussion section- adding references to support findings.

3. Include statistical analysis to support and validate the quantitative part.

4. The text must be improved in English.

Reviewer #5: Manuscript Title: Exploring Physical Activity and Patient Perceptions in Knee Osteoarthritis: A Mixed-Methods Study

Manuscript ID: PONE-D-25-20426

An excellent sequential explanatory mixed-methods study of physical activity levels and patient opinions of knee osteoarthritis (OA) management in Saudi Arabia is presented in this publication. Because it focuses on a Middle Eastern population, the study is methodologically sound, ethically appropriate, and contextually meaningful. Both quantitative and qualitative data are well integrated. Small changes are required to improve data openness, clarify methods, and focus the conversation.

Introduction:

1. Novelty could be more clearly distinguished from previous mixed-methods studies in OA.

Methodology:

1. The cutoff for “low activity” (≤438 MET-min/week) using IPAQ-SF is non-standard. Please clarify and cite validation studies for this threshold.

2. A sample size justification for the quantitative phase (n=60) is missing.

Results and Data Presentation

1. Table formatting (Table 1, Table 2) should be cleaned for clarity and consistency.

2. Avoid duplicating large blocks of quotes in both Table 3 and the Results section – summarize selectively.

Discussion:

1. Potential gender-based insights, given equal male/female participation, are not discussed.

Data Availability

Current statement ("data available on request") does not meet PLOS ONE’s data availability policy. Authors must deposit data in a public repository or provide detailed justification for restrictions.

Reference:

Ensure full and consistent formatting (some DOIs and titles are not uniform).

Reviewer #6: The study addresses an important gap in understanding physical activity levels and patient perceptions in knee osteoarthritis (OA), particularly in the Saudi Arabian context. The mixed-methods approach is well-justified and enhances the depth of findings. However there are some points needed to modify: 1.Self-Report Bias:

The reliance on self-reported physical activity (IPAQ-sf) may introduce recall bias.

- Suggestion: Triangulate with objective measures (e.g., accelerometers) in future research. Lack of Longitudinal Data The cross-sectional design prevents causal inferences about barriers and activity levels.

- Suggestion: Propose longitudinal studies to track changes over time.

2.Thematic Analysis Depth:

While themes are well-identified, deeper exploration of subthemes (e.g., specific misconceptions about OA) could enrich the qualitative findings.

- Suggestion: Include more direct quotes to illustrate nuances.

3.Data Availability:

The statement "Data available on request from the author" is insufficient for reproducibility.

- Suggestion: Deposit anonymized data in a public repository or provide a detailed access protocol.

4.Structural Clarity:

- The "Discussion" could better align themes with existing literature (e.g., compare findings to global studies on OA barriers).

- Suggestion: Reorganize the discussion to first recap key findings, then compare to prior work, and finally state implications.

Grammar/Clarity:

- Page 8, Line 15: "Physical activity is a foundation of osteoarthritis management" → "Physical activity is foundational to osteoarthritis management."

- Page 26, Line 422: "Integrating PA-level assessment with qualitative insights offers a robust understanding" → "Integrating physical activity assessments with qualitative insights provides a robust understanding."

Reviewer #7: Comments:

1-The conclusions presented in both the abstract and main text suggest that tailored interventions can “significantly improve” adherence to physical activity and “overall quality of life.” While the qualitative findings do highlight perceived barriers and facilitators, no outcome data (e.g., adherence, functional gains, or quality of life) were directly measured in the study. These statements should be revised to reflect the study's actual scope—i.e., exploring perceptions and reported physical activity levels—rather than implying causal or outcome-based effects.

2-The term “low physical activity” is appropriately defined using IPAQ thresholds, but the phrase “critically low” in the abstract and discussion requires clarification. It suggests a clinical severity or risk threshold, yet the study does not report on associated health outcomes. The term should either be clearly defined or replaced with a more neutral description.

Similarly, the use of “tailored interventions” throughout the manuscript implies a level of design or testing that is not part of this study. It should be clear that such interventions are recommended based on findings, not assessed directly.

3- tthe manuscript occasionally uses phrases such as “evaluate physical activity levels and patient experiences,” which may be misread as a longitudinal or intervention study. Clarifying that this is a cross-sectional, exploratory, perception-based study would improve accuracy.

4-Demographic findings, such as BMI, education level, and gender, are presented in the Results section but are not meaningfully discussed in relation to the study’s main findings. For example, does BMI correlate with perceptions of mobility barriers? Did educational level affect awareness of physical therapy benefits? If these data were not explored, the demographic summary should be clearly described as contextual rather than implying analytic weight.

5-The discussion emphasizes “general” implications for OA care, but the study is conducted in a single country with a culturally specific population. The findings around prayer, family involvement, and community misconceptions are highly context-bound. The authors should temper generalizations and clarify that these insights may not be transferable to other settings without cultural adaptation.

6-The concluding statement—“addressing barriers through tailored interventions…can significantly improve adherence and quality of life”—goes beyond what this descriptive study supports. No interventions were tested, nor were changes in quality of life measured.

Example of alternative: “Our findings suggest that addressing barriers through culturally sensitive, patient-centred strategies may support increased physical activity engagement among individuals with knee OA.”

7- The final sentence would benefit from clearer structure. Suggest: “Our findings underscore the importance of addressing these barriers through patient-centred education and improved access to physical therapy.”

8- Methods: Please clarify how saturation was determined in the qualitative study. Was a formal stopping rule or documentation strategy used?

9- This manuscript provides important insights into perceived barriers to physical activity among individuals with knee OA in Saudi Arabia using a robust mixed-methods design. However, the conclusions currently overstate the implications of the findings, and clarification is needed around terminology, scope, and potential biases.

10- The manuscript should clearly state that the “perceptions and awareness” were only qualitatively assessed.

**Do you want your identity to be public for this peer review?** For information about this choice, including consent withdrawal, please see our Privacy Policy

Reviewer #1: No

Reviewer #2: **Yes: ** Ahmed Ibrahim Al Kharusi

Reviewer #3: No

Reviewer #4: **Yes: ** Dan Iulian Alexe

Reviewer #5: No

Reviewer #6: No

Reviewer #7: **Yes: ** Fawaz Alrasheedi

---

## [Author Response · Author response to Decision Letter 1]

24 Sep 2025

We sincerely thank the Academic Editor and Reviewers for their constructive feedback. We have carefully revised our manuscript in response to all comments. Attached a detailed, point-by-point response, with changes highlighted in the revised manuscript and page/line numbers indicated.

---

## [Decision Letter · Decision Letter 1]

29 Oct 2025

Exploring Physical Activity and Patient Perceptions in Knee Osteoarthritis: A Mixed-Methods Study

PONE-D-25-20426R1

Dear Dr. Subahi,

We’re pleased to inform you that your manuscript has been judged scientifically suitable for publication and will be formally accepted for publication once it meets all outstanding technical requirements.

Kind regards,

Domiziano Tarantino, MD

Academic Editor

PLOS ONE

Additional Editor Comments (optional):

Reviewers' comments:

Reviewer's Responses to Questions

**Comments to the Author**

Reviewer #1: All comments have been addressed

Reviewer #3: (No Response)

Reviewer #4: All comments have been addressed

Reviewer #6: All comments have been addressed

2. Is the manuscript technically sound, and do the data support the conclusions?

Reviewer #1: Yes

Reviewer #3: (No Response)

Reviewer #4: Yes

Reviewer #6: Yes

3. Has the statistical analysis been performed appropriately and rigorously?

Reviewer #1: Yes

Reviewer #3: (No Response)

Reviewer #4: Yes

Reviewer #6: Yes

4. Have the authors made all data underlying the findings in their manuscript fully available?

Reviewer #1: Yes

Reviewer #3: (No Response)

Reviewer #4: Yes

Reviewer #6: Yes

5. Is the manuscript presented in an intelligible fashion and written in standard English?

Reviewer #1: Yes

Reviewer #3: (No Response)

Reviewer #4: Yes

Reviewer #6: Yes

Reviewer #1: The authors have satisfactorily addressed all previous comments and provided clear revisions throughout the manuscript. The responses are comprehensive, and the updated version has improved clarity and flow. I have no further concerns and believe the manuscript is suitable for publication in its current form.

Reviewer #3: In order to check the application of reviewers' opinions, Changes made are highlighted or made with track changes.

Reviewer #4: I have no further comments. Thank you to the authors for their effort in responding to my recommendations and observations.

Good luck

Reviewer #6: All comments have been addressed abd if the other reviewers accept the paper, it's ready for publication l.

**Do you want your identity to be public for this peer review?** For information about this choice, including consent withdrawal, please see our Privacy Policy

Reviewer #1: No

Reviewer #3: No

Reviewer #4: **Yes: ** Dan Iulian Alexe

Reviewer #6: No

---

## [Editor Report · Acceptance letter]

PONE-D-25-20426R1

PLOS ONE

Dear Dr. Subahi,

I'm pleased to inform you that your manuscript has been deemed suitable for publication in PLOS ONE. Congratulations! Your manuscript is now being handed over to our production team.

Kind regards,

on behalf of

Dr. Domiziano Tarantino

Academic Editor

PLOS ONE